# Serum Proteins Associated with Blood–Brain Barrier as Potential Biomarkers for Seizure Prediction

**DOI:** 10.3390/ijms232314712

**Published:** 2022-11-25

**Authors:** Elżbieta Bronisz, Agnieszka Cudna, Aleksandra Wierzbicka, Iwona Kurkowska-Jastrzębska

**Affiliations:** 1Second Department of Neurology, Institute of Psychiatry and Neurology, 02-957 Warsaw, Poland; 2Sleep Disorders Center, Department of Clinical Neurophysiology, Institute of Psychiatry and Neurology, 02-957 Warsaw, Poland

**Keywords:** MMP-2, MMP-9, CCL-2, blood–brain barrier, inflammation, epilepsy, seizure prediction, predictive models

## Abstract

As 30% of epileptic patients remain drug-resistant, seizure prediction is vital. Induction of epileptic seizure is a complex process that can depend on factors such as intrinsic neuronal excitability, changes in extracellular ion concentration, glial cell activity, presence of inflammation and activation of the blood–brain barrier (BBB). In this study, we aimed to assess if levels of serum proteins associated with BBB can predict seizures. Serum levels of MMP-9, MMP-2, TIMP-1, TIMP-2, S100B, CCL-2, ICAM-1, P-selectin, and TSP-2 were examined in a group of 49 patients with epilepsy who were seizure-free for a minimum of seven days and measured by ELISA. The examination was repeated after 12 months. An extensive medical history was taken, and patients were subjected to a follow-up, including a detailed history of seizures. Serum levels of MMP-2, MMP-9, TIMP-1, CCL-2, and P-selectin differed between the two time points (*p* < 0.0001, *p* < 0.0001, *p* < 0.0001, *p* < 0.0001, *p* = 0.0035, respectively). General linear model analyses determined the predictors of seizures. Levels of MMP-2, MMP-9, and CCL-2 were found to influence seizure count in 1, 3, 6, and 12 months of observation. Serum levels of MMP-2, MMP-9, and CCL-2 may be considered potential biomarkers for seizure prediction and may indicate BBB activation.

## 1. Introduction

Epilepsy is a chronic disease affecting the central nervous system (CNS) and characterized by the occurrence of seizures. Of the 1% of the world population who are affected by epilepsy, 30% are drug-resistant. Drug-resistant epilepsy is defined as “a failure of adequate trials of two tolerated, appropriately chosen and used antiepileptic drug schedules (whether as monotherapies or in combination) to achieve sustained seizure freedom” [1]. For these patients, research involving new antiseizure and antiepileptic drugs is crucial, yet with over 30 drugs already on the market, the percentage of drug-resistant patients has not changed. Therefore, it is essential to develop biomarkers, i.e., “characteristics that are objectively measured and evaluated as an indicator of normal biological processes, pathogenic processes, or pharmacological responses to therapeutic intervention” [2], that can predict seizure in these patients.

BBB dysfunction and neuroinflammation are factors underlying seizure induction and epileptogenesis. Pathogens, blood cells, and endo- and exogenic substances enter the brain through compromised BBB leading to local inflammation and angiogenesis, which may exacerbate epilepsy [3]. Interactions between leukocytes and endothelium and leukocyte influx to the brain are also important elements of epileptogenesis [4,5]. BBB dysfunction coexists with neuroinflammation, which even further increases BBB permeability [6,7]. While seizures induce an inflammatory response in reaction to hypoxia, reactive oxygen species, calcium, and glutamate, inflammation and BBB activation are associated with the exacerbation of seizures [3,4,8,9,10].

Numerous molecules and cells are involved in the regulation of BBB homeostasis and inflammation. Matrix metalloproteinases (MMPs) impact ECM composition, influence BBB permeability, and modulate inflammatory response [11,12]. Expression of MMP-2 and MMP-9 increases in the brain of patients with epilepsy [13], and serum levels of MMP-9 are elevated after seizures [14]. Tissue inhibitors of metalloproteinases (TIMPs) act as a counterbalance to MMPs, and their overexpression is correlated with the preservation of BBB [15]. Levels of TIMP-1 and TIMP-2 are elevated after generalized tonic-clonic seizures [16]. TSP-2 can act as a non-specific inhibitor of MMPs. In a model of brain injury, loss of TSP-2 was associated with exacerbation of inflammation and slowing of BBB repair [17]. In patients with TLE who were undergoing epilepsy surgery, serum levels of TSP-2 were higher in comparison to the control group [18].

Adhesion molecules such as ICAM-1 are essential for leukocyte binding. Leukocyte binding results in functional changes in endothelial cells leading to their shrinking, loosening of chemical bonds of tight junctions, and facilitation of leukocyte transmigration through BBB [19]. P-selectin participates in leukocyte rolling [20]. In patients with status epilepticus and comorbidities, serum ICAM-1 level was increased [21], and in animal models, expression of P-selectin was increased after seizures [4]. CCL-2 is a chemokine participating in leukocyte adhesion [22] and exacerbating inflammation [23], whose expression increases after seizures [24]. Serum S100B correlates with the albumin quotient and is considered an indirect marker of neuronal damage [25]. Serum level of S100B increases after seizures [16,26], indicating impairment of the BBB.

In this study, we chose molecules whose levels are not only known to be affected in epilepsy but also which can be measured in serum (MMP-9, MMP-2, CCL-2, S100B, TIMP-1, TIMP-2, ICAM-1, TSP-2, P-selectin) [4,9,11,18,27,28]. The aim of the study was to assess if the examined molecules can be considered biomarkers of seizure prediction. We hypothesized that higher levels of the molecules increasing BBB permeability and inflammation (MMP-9, MMP-2, CCL-2, S100B, ICAM-1, P-selectin) would result in a higher number of seizures while higher levels of the molecules improving BBB integrity (TIMP-1, TIMP-2, TSP-2) will result in lower seizure count. 

## 2. Results

The research group consisted primarily of 50 patients, of which 1 patient was excluded due to lack of follow-up blood sampling, which yielded 49 patients, 29 women and 20 men (Appendix A). All 49 patients completed follow-ups and were analyzed in the study. The group consisted of 18 patients with idiopathic generalized epilepsy, 24 patients with structural-metabolic epilepsy, and 7 patients with unknown etiology. The clinical and demographic characteristics of the patients are summarized in Table 1, Table 2, Table 3, Table 4, Table 5, Table 6, Table 7 and Table 8.

The patients recorded seizures in a seizure diary on a weekly basis. During one year of follow-up, the disease course in the observed patients was stable. The mean number of seizures was 7.31 ± 2.54, and the median was 0, with a maximum seizure count of 108. There were no statistically significant differences in seizure count during one year of follow-up and seizure count recorded during one year before baseline (mean 6.18 ± 1.56, median 1, min–max 0–57). In the examined group of patients, there were no additional diseases, traumas, status epilepticus, or any other medical events belonging to exclusion criteria that occurred during the follow-up. Treatment was changed in 21 patients, including 18 patients with dosage change and 3 patients with a medication change.

We evaluated the levels of the molecules at baseline and follow-up and found that the levels of the majority of the markers significantly differed even though the epilepsy course remained similar (Table 9). Levels of MMP-9, TIMP-1, MMP-2, and MMP-2/TIMP-2 ratio were higher at baseline, while the levels of CCL-2 and P-selectin were higher at follow-up. Levels of S100B, TIMP-2, TSP-2, and ICAM-1 and MMP-9/TIMP-1 ratio were similar.

To explain the differences in the levels of the examined molecules, we checked if there were any changes in the epilepsy course. There were no significant differences in seizure count and seizure severity at baseline and at follow-up (Appendix A), and no significant medical events which could influence the levels of the biomarkers.

We examined correlations between seizure count before the examination, measured within the range of 12, 6, 3, and 1 month, and serum levels of the examined proteins and found that there were no significant correlations between these parameters. Out of the panel of the markers, only TIMP-2 showed a weak negative correlation with seizure count (at 12 months: r = (−)0.25, *p* = 0.0127; 6 months: r = (−)0.25, *p* = 0.0123; 3 months: r = (−)0.31, *p* = 0.0015; 1 month: r = (−)0.29; *p* = 0.0031).

There were no statistically significant differences between the patients with normal EEG activity and patients with EEG abnormalities (abnormal background activity, slow waves, generalized/focal abnormality, epileptiform discharges). There were no statistically significant differences in levels of molecules in patients treated with monotherapy and polytherapy. There was no correlation between the 2- and 5-year risk recurrence and 10-year chance of seizure freedom obtained from the EPT calculator and levels of examined molecules.

To exclude the influence of the changes in medication regimen on serum protein levels, we checked for differences between the group of patients with treatment changes and the group of patients with no changes in treatment. The levels of the majority of the molecules did not differ with the exception of MMP-9, which was higher in the group of patients with treatment change (mean 464.05 ± 128.99 ng/mL vs. 212.43 ± 32.48 ng/mL, *p* = 0.0322).

Additionally, we compared patients with seizures and patients without seizures as recorded in history and follow-up. Out of the 49 patients, 14 did not have seizures during the year preceding the examination and the year of follow-up, while 35 patients had at least one seizure. There were no significant differences in the levels of biomarkers between the no-seizure and seizure groups at baseline (Appendix A). However, in the group without seizures, the levels of most of the examined molecules (MMP-9, S100B, ICAM-1, P-selectin, MMP-2, TIMP-2, TSP-2) were stable between baseline and follow-up (Appendix A) while in the subgroup experiencing seizures only levels of S100B and TSP-2, and MMP-9/TIMP-1 ratio was stable (Appendix A).

GLMMs were used to identify and evaluate the importance of various factors in predicting seizure count after the examination. AIC statistics were used for model selection. Variables with many zero values were analyzed using the Tweedie distribution. In the GLMMs, we found that levels of MMP-9, MMP-2, and CCL-2 influenced seizure count after the examination. MMP-9 and CCL-2 were observed to influence the seizure count at seven days, 1 month, 3, 6, and 12 months of follow-up. MMP-2 influenced seizure count at 1 month, 3, 6, and 12 months of follow-up (Figure 1, Appendix A). Higher levels of MMP-2 and CCL-2 were associated with lower seizure counts, whereas a higher level of MMP-9 was associated with higher seizure counts. The models that best described the prediction of seizures included co-variables, time of remission, disease duration, seizure count during the last three months, and sex. A longer time of remission, shorter duration of epilepsy, lower number of seizures during the last three months, and female sex were associated with lower seizure count. The preliminary finding of the weak positive correlation between serum P-selectin and seizure count was not confirmed in the GLMMs.

## 3. Discussion

Numerous studies have addressed the link between BBB dysfunction, local CNS inflammation, and seizures; however, the majority of the studies carried out in patients focused on the alterations in CSF and serum molecule levels in a short period of time after seizures [14,26]. Few studies examined the correlation of BBB-associated serum molecules with epilepsy diagnosis [29,30,31], and even fewer investigated them as biomarkers of seizure prediction [21,32,33].

Our study analyzed the potential of molecules associated with the regulation of BBB function and neuroinflammation as biomarkers of seizure prediction. We observed the influence of MMP-9, MMP-2, and CCL-2 on seizure count. Surprisingly, the levels of matrix metalloproteinases (MMPs) did not correlate with seizure count in the same direction–a higher level of MMP-9 was associated with higher seizure count as was hypothesized, but a higher level of MMP-2 was observed in patients with lower seizure count. A serum level of CCL-2 influenced seizure count similarly to MMP-2.

Another important finding was that the serum levels of the MMP-9, TIMP-1, MMP-2, CCL-2, and P-selectin and MMP-2/TIMP-2 ratio differed, even though the course of disease measured by seizure count was stable. At the same time, levels of S100B, TIMP-2, TSP-2, ICAM-1, and MMP 9/TIMP-1 ratio remained unchanged. Of those, serum levels of S100B, TSP-2, and MMP 9/TIMP 1 ratio were stable even in the subgroup of 35 patients who experienced seizures. A change in medication regimen was associated with a higher level of MMP-9.

### 3.1. MMP-9

In our study, MMP-9 was shown to influence seizure count and, as such, could be considered a potential biomarker of seizure prediction. MMP-9 belongs to the family of MMPs, molecules playing a major role in many physiological and pathological processes. MMPs participate in cell differentiation and migration, tissue remodeling and repair, cytokine secretion, and regulation of homeostasis between pro- and anti-inflammatory factors, but also neuroinflammation, neurotoxicity, and carcinogenesis [11,34,35]. MMP-9 is a specific member of this family, being a complex molecule with the ability to bind with numerous substrates, such as TIMPs, collagen type I and IV, tight junction proteins, interleukins, growth factors, and adhesion molecules [36]. MMP-9 is primarily produced by activated microglia and astrocytes [37]. In the neural and glial cells, it is localized in the nucleus [38,39]. A small amount of MMP-9 is secreted constitutively, and its expression rises significantly in reaction to inflammation, neuronal depolarization, or activation of various neuronal receptors [40,41,42,43]. Usually, MMP-9 is secreted in a complex with TIMP-1 [44], but it can be released in a free form by neutrophils [45].

MMP-9 affects the induction of seizures and epileptogenesis by complex changes of both environment and structure of neurons, causing lysis of basal lamina and ZO-1 protein [46], affecting cytokine release from the extracellular matrix (ECM) and their activation [12], facilitating leukocyte migration through ECM [47], increasing activity of NMDAR [48], decreasing activity of AMPAR [49], influencing the morphology of dendritic spines and reorganization of neuronal circuits [49], and contributing to neuronal cell death [50]. Serum level of MMP-9 is increased after tonic-clonic seizures (TCS) at 30 min, 2, 6, and 24 h and returns to normal at 72 h [14,26]. 

MMP-9 is known to increase the susceptibility to kindling in animal models [38], which facilitates epileptogenesis and seizure induction. In the study on seizure prediction, serum levels of IL 6, IFN-γ, I IFN-δ3, and IL-17a were correlated with time to the next seizure episode [51]. Out of those inflammatory cytokines, IL-6 and IL-17a increase MMP-9 expression outside CNS [52,53]. It seems highly probable that they behave in a similar way in the brain, increasing MMP-9 expression and, as a result, leading to seizure induction. In our study higher level of MMP-9 in patients with treatment change might indicate a more active course of disease in these patients, which also suggests an impact of MMP-9 on disease activity.

The observed influence of serum MMP-9 on seizure count might suggest that this molecule contributes to seizure generation in patients with epilepsy, yet it is still unclear which of its mechanisms of action prevail and if the serum level of MMP-9 reflects MMP-9 activity in the CNS. With those limitations, serum MMP-9 in patients with epilepsy might be considered a biomarker of seizure prediction.

### 3.2. MMP-2

MMP-2 is another member of the family of MMPs whose influence on seizure count was shown in the present study. MMP-2 is a molecule co-acting with MMP-9 in the degradation of basal lamina and tight junction proteins [54], leukocyte migration, activation of interleukins and growth factors [55], increasing chemokine expression [56], and modification of dendritic spines [57]. MMP-2 is secreted by astrocytes, endothelial cells, pericytes, and blood-derived macrophages [58,59,60,61]. It is secreted mostly in a constitutive manner [62], but inflammation and neurotoxic substances can increase its secretion [61,63]. 

In a rat model of epileptogenesis induced by electrical stimulation, expression of MMP-2 was increased in CA3 and entorhinal cortex at seven days after status epilepticus [64]. Increased expression of MMP-2 was found in the brains of adult patients with focal cortical dysplasia (FCD) and hippocampal sclerosis (HS) [13,65]. However, the serum level of MMP-2 after seizures was not significantly increased [16], and in a population of epilepsy patients with a mean time of remission of 164 days, it was lower than in controls [29]. In our study lower serum level of MMP-2 was associated with a higher seizure count at 1 month, 3, 6, and 12 months of follow-up, which might suggest that the amount of the molecule that is used during seizures exceeds the amount that is secreted. Therefore, a lower serum level of MMP-2 might indicate higher disease activity, yet further research is needed to explain the exact mechanisms. With precautions, serum MMP-2 might be considered a potential biomarker of seizure prediction.

### 3.3. CCL-2

Serum level of CCL-2 influenced seizure count similar to the serum level of MMP-2–lower level of CCL-2 was associated with higher seizure count at seven days, 1 month, 3, 6, and 12 months of follow-up. CCL-2, earlier known as MCP-1, is a molecule belonging to the family of chemokines. It is a proinflammatory mediator, acting as a chemoattractant for monocytes, lymphocytes T and NK cells, participating in leukocyte adhesion by activation of integrins [22], activating leukocytes and microglia [66], modifying ECM [67] and tight junction proteins [68], and therefore, influencing BBB permeability. CCL-2 exacerbates inflammation [23], impairs Ca2+ buffering [69], and modifies the expression of synaptic proteins [70]. Its overexpression resulted in aberrant neuronal plasticity in the mouse hippocampus [71].

In the CNS, CCL-2 is secreted by astrocytes, neurons, microglia, endothelial cells, and blood-derived leukocytes [72,73,74,75]. The molecule is secreted constitutively at low concentrations [76], and its expression is increased after exposure to proinflammatory mediators [72,75]. In animal models, expression of CCL-2 increased after status epilepticus [24] and in a model of systemic inflammation with seizures [77]. Inhibition of CCL-2 transcription and administration of CCR2 antagonist or CCL-2 neutralizing antibodies resulted in lower seizure count [77]. CCL-2 was also overexpressed in specimens from patients with mesial temporal lobe epilepsy (MTLE), HS, and FCD type II [78,79,80]. Serum level of CCL-2 in patients was observed to be either increased [81] or did not differ from controls [82,83].

In the present study, a lower serum level of CCL-2 was associated with a higher number of seizures. As CCL-2 acts as an activator of microglia and leukocytes within the brain and intensifies local inflammation by attracting further immune cells and facilitating leukocyte adhesion, it is a molecule greatly involved in local neuroinflammation. CCL-2 binds its key receptor, CCR2, localized within the CNS in activated microglia and macrophages [84,85]. Activation of the CCL-2/CCR2 pathway was found to be responsible for the pro-inflammatory and pro-seizure activity of CCL-2 [77]. Additionally, expression of both CCL-2 and CCR2 increases after seizures [86]. It is, therefore, possible that the lower level of serum CCL-2 that was observed to be associated with higher disease activity in our study reflects a low level of unbound CCL-2 which remained available for analysis in serum, while there might have been an amount of CCL-2 that formed a complex with CCR2, activated CCL-2/CCR2 pathway in the CNS, and contributed to the induction of seizures. The exact mechanisms are yet to be unveiled. With these limitations, the serum level of CCL-2 might be considered a candidate for a biomarker of seizure prediction.

### 3.4. Other Molecules

Serum levels of TIMP-1, TIMP-2, S100B, P-selectin, ICAM-1, and TSP-2 did not influence seizure count in the GLMMs in the period of seven days, 1, 3, 6, and 12 months.

Disbalance between MMPs and TIMPs is associated with many CNS diseases. Apart from inhibiting MMPs, TIMPs influence cell growth [87], migration and apoptosis [88], and activation of growth factors [89]. In our study, serum levels of MMP-9 and MMP-2 influenced seizure count, yet no such correlation was observed for TIMP-1, TIMP-2, and a non-specific inhibitor of MMPs, TSP-2. The observed lack of influence might suggest the prevalence of importance of BBB-impairing molecules (MMP-2, MMP-9) over the molecules playing a neuroprotective role (TIMP-1, TIMP-2, TSP-2) in seizure induction.

Serum levels of P-selectin and ICAM-1, molecules associated with leukocyte trafficking, did not influence seizure count. The stability of the molecule levels can reflect the stability of the processes of BBB activation within the brain, yet the results need further research as serum P-selectin is known to increase no more than fourfold over the control group [90] and soluble ICAM-1 can be produced not only by endothelial cells but also aortic smooth muscle cells and hematopoietic cell lines [19].

In our study, the serum level of S100B did not influence seizure count at follow-up and did not differ between baseline and follow-up. S100B may influence pathways leading to both neuroprotection and neurotoxicity [91]. At low concentrations, S100B is associated with nerve growth [92], long-term potentiation (LTP) [93], neuronal survival [94], and degradation of radical oxygen species [95]. High concentration of the molecule correlates with toxicity and inflammation [95,96]. In the present study, we found that serum S100B level does not influence seizure count. It might be associated with the overall low level of S100B in our patients. According to Marchi et al., concentrations of S100B below 340 pg/mL might indicate increased BBB permeability without damage to the CNS [97]. Therefore, S100B could act more as a neuroprotective than a neurotoxic agent in our patients, yet it is also plausible that in a group of epilepsy patients with higher activity of disease (i.e., more frequent and/or more severe seizures), the level of S100B would differ and S100B would contribute to neuronal damage and intensification of inflammation.

### 3.5. Co-Variables

In our study, the GLMMs for seizure prediction showed not only molecule levels but also other co-variables significantly influencing seizure count: time of remission, disease duration, seizure count during the last three months, and sex. Our data imply that a longer time of remission is a predictor of lower seizure count, which replicates many previous findings [98,99]. In the study of Lamberink et al. longer time of remission was associated with a lower risk of seizure recurrence at two and five years and a higher ten-year chance of seizure freedom [99]. In our study longer duration of epilepsy was associated with a higher seizure count. A similar conclusion comes from the abovementioned study in which a longer period of disease duration counted from the occurrence of the first to the last seizure is associated with a higher lower risk of seizure recurrence at two and five years and a higher ten-year chance of seizure freedom [99] and from the study assessing prognostic factors in patients with epilepsy who underwent surgery due to HS or neurocysticercosis [100]. Male sex was associated with a higher seizure count, a result also obtained by other authors [101] but not consistent in the literature [102,103]. A higher number of seizures during the last three months was associated with a higher seizure count during follow-up. Similarly, in a recent study, seizure frequency below one seizure/month was a predictor of epilepsy remission [102]. 

EEG did not influence epilepsy prognosis in our patients. This result might be associated with a low number of patients with epileptiform discharges (*n* = 9) in our group. EEG abnormalities have long been recognized as predictors of worse outcomes in epilepsy [99,102,104], yet the predictive value of epileptiform abnormalities is not consistent between the studies [105]. The calculated risk of seizure recurrence at two and five years and ten-year chance of seizure freedom [99] did not correlate with serum levels of the examined molecules. This might be due to long periods (two, five, and ten years) from the time of assessment and thus the too low influence of the molecule levels on seizures, yet as newer studies reassess the calculator of seizure recurrence risk [105], there is also the possibility that this tool would be refined. There were no statistically significant differences in levels of molecules in patients treated with monotherapy and polytherapy, which suggests that different treatment regimens do not influence the serum levels of the examined molecules.

### 3.6. Strengths and Limitations

The utility of serological markers for epilepsy prognosis is under-researched, and our study tries to fill in this scientific gap. Prediction of seizures remains the main problem in patients with uncontrolled seizures, and as 30% of epilepsy patients remain drug-resistant, seizure prediction is of major importance. Therefore, our study deals with an important problem affecting a large group of people. In patients with uncontrolled epilepsy, seizure prediction can impact both small daily decisions and changes in treatment. Implementing biomarkers of seizure prediction could therefore be of great value for the patients and their caregivers. Another strong point of the study is the use of specifically designed GLMMs that identify co-variables influencing seizure count at different periods of follow-up.

The main limitations of the study are that the presented data is observational and the sample size is small. Hence, the results need to be considered with care and in comparison with other research, which is why we discussed the data comparatively. The proteins we had chosen have a clinical correlation with neuroinflammation, endothelium, and blood–brain barrier activation, yet, their range of actions is wide, and they participate in many more physiological and pathological processes. To minimize the influence of these processes, we used a very specific set of exclusive criteria. 

To avoid the potential influence of circadian rhythm on the molecule levels (not known for all of the evaluated molecules, but already shown for CCL-2 and ICAM-1 [106] and as an inflammatory response is circadian based what is likely to affect levels of MMP-9, MMP-2, TIMP-1, S100B, and P-selectin) we took blood samples only in the morning. As the half-life of some of the examined proteins is known to be short (e.g., S100B [107], CCL 2 [108]), we took great care in the fast processing of blood samples. To minimize the effect of physical exertion, the patients were asked to refrain from demanding physical activity 24 h before blood collection.

The potential bias of the research may involve a specific group of epilepsy patients who participated in the study. Even though the patients were enrolled on the basis of the consecutiveness of their appointments in the outpatient clinic, the clinic itself is a part of the Institute of Psychiatry and Neurology, which is a tertiary referral hospital. Therefore, part of the patients who attended the outpatient clinic were hospitalized before due to epilepsy in the Neurological Departments, and as such, the course of the disease in these patients may be more severe than in the general population. Additionally, the outpatient clinic deals specifically with patients with epilepsy, and it may happen that patients with a milder course of the disease are not referred to the specific epilepsy outpatient clinic but are treated by general neurologists. Thus, it is possible that patients enrolled in the present study have higher seizure counts and greater seizure severity and take more medication than the general population of epilepsy patients.

The task of seizure prediction is also hampered by the fact that serum levels of MMP-9, MMP-2, CCL-2, P-selectin, and TIMP-1 differed between the two time points, even though the seizure count and seizure severity were similar. The changing level of the molecules suggests that they are influenced not only by the occurrence of seizures. As inflammation, neurodegeneration, and BBB activation occur in the brain of patients with epilepsy, there is more than one possible answer to the question of why the level of these molecules might change. Apart from the presence of peripheral factors, we tried to exclude them during the study. The differences in serum levels of MMP-9, MMP-2, CCL-2, P-selectin, and TIMP-1 also indicate that multiple blood sampling should be carried out to ensure reliable results of the analysis.

The statistical analysis involved the models from the class of GLMMs, and as with every statistical method, the GLMMs have their limitations [109]. In light of these limitations, we carefully selected models from the class of GLMMs using the practical information-theoretic approach [110] to avoid misselection. A more complex discussion on the limitations of the GLMMs is beyond the scope of this article, yet it could be found in [109].

## 4. Materials and Methods

Fifty successive adult patients with epilepsy attending the outpatient clinic of the Institute of Psychiatry and Neurology who fulfilled inclusion criteria did not fulfill exclusion criteria, and confirmed their willingness to participate in the study by signing informed consent were included in the study in the period from 14 October 2015 to 16 May 2017. 

The inclusion criteria were diagnosis of epilepsy (on the basis of ILAE criteria from 2014 [111]) and a minimum time of seizure freedom of seven days. Duration of seizure-free period was based on our previous studies [14]. The exclusion criteria included every state or disease which was known to influence the examined markers (malignant tumor, inflammatory disease, severe ongoing neurological disease with vascular damage (acute stroke within last 6 months), neuroimmunological disease, history of immunosuppressive or immunomodulatory treatment during last 6 months, surgery or significant trauma within last two weeks, hepatic, renal or cardiac insufficiency, severe psychiatric disease, symptoms of infection or CRP above the laboratory norm, pregnancy). Number of seizures did not affect recruitment to the study.

The patients’ demographic data and extensive medical history, including disease etiology, type of seizures, age at onset, disease duration, treatment regimen, time of remission, and seizure count in the period of last 1, 3, 6, and 12 months were taken at baseline (T0). Patients were assigned to different etiologic groups (genetic, structural-metabolic, unknown) based on the classification proposed by ILAE in 2010 [112]. Seizure severity was assessed with National Hospital Seizure Severity Scale (NHS3 [113]). The risk of seizure recurrence at 2 and 5 years and 10-year chance of seizure freedom was assessed with Epilepsy Prediction Tool (EPT [99]).

Patients were advised to fast before the blood collection and refrain from smoking, using alcohol, and undertaking strenuous physical activities 24 h before blood collection. Blood was collected in the morning between 8 and 9 a.m. after 5 to 15 min rest to ascertain similar conditions of blood collection. Samples were centrifuged, and the obtained supernatant was divided into two parts: one part was frozen and stored in closed test tubes at −80 °C for further analysis, and the second part was immediately examined using standard laboratory tests for CRP. After blood collection, we performed a standard EEG examination in 10–20 montages for 30 min. 

Patients were followed up for two years and were obliged to keep a weekly seizure diary (checked every 3 months) and inform the researchers of any events which could influence the level of the measured molecules (as listed above in exclusion criteria). After one year of follow-up (T1), blood was collected for the second time according to the same regime as at the first blood collection. The follow-up blood collection (T1) took place from 12 October 2016 to 17 May 2018. The samples were handled as previously described. Total period of follow-up (until T2) lasted until 17 May 2019.

The frozen serum was thawed prior to the immunoenzymatic testing. The analysis of 9 markers of BBB activation and/or inflammation (MMP-9, MMP-2, TIMP-1, TIMP-2, S100B, TSP-2, CCL-2, ICAM-1, P-selectin) was performed on serum samples using sandwich-type ELISA kits following manufacturers’ instructions. The producer of MMP-9, MMP-2, TIMP-1, TIMP-2, TSP-2, CCL-2, ICAM-1, and P-selectin kits was R&D Systems, Minneapolis, MN, USA, the producer of S100B kit was Merck Millipore, Darmstadt, Germany. Data from the reactions was acquired using the Multiskan Go microplate reader (Thermo Scientific, Waltham, MA, USA), and the concentration was calculated as advised by the manufacturers.

The data was preliminarily analyzed with descriptive statistics such as means, standard deviation, percentiles, and correlations. One-dimensional calculations for continuous variables were performed using Student’s or Wilcoxon’s tests depending on the probability distributions. Nominal variables were summarized in contingency tables and analyzed using the Chi-square test or Fisher’s exact test. The GLMMs (Generalized Linear Mixed Models) were used to identify and evaluate the importance of various factors in predicting seizure count after the examination. AIC (Akaike Information Criterion) statistics were used for the optimal model selection as AIC statistic is asymptotically equivalent to resampling methods [114]. Models were selected using the practical information-theoretic approach [110]. Most of the variables were modeled with the assumption of a normal distribution; in some cases, the gamma distribution was optimal. Additionally, variables with many zero values (over 10%) were analyzed using the Tweedie distribution. The calculations were performed in the SAS package (SAS/STAT rel. 15.1). The p value 0.05 was taken as the significant confidence level. There was no missing data. The data analysis was performed from 5 May 2016 to 13 August 2021. 

The manuscript was prepared according to the STROBE checklist [115]. The timeline diagram is depicted in Figure 2.

## 5. Conclusions

The search for epilepsy biomarkers, both diagnostic and prognostic, is still ongoing. Biomarkers could help individualize the treatment of epilepsy according to epilepsy etiology and predict seizure count and severity. Our study addresses the somewhat neglected field of serum molecules that can be used to predict seizures and plan epilepsy treatment. As there are no ideal peripheral biomarkers, we think that serum molecules associated with the blood–brain barrier and neuroinflammation could be “close to ideal” biomarkers predicting seizures—not demanding wearing any mechanical devices, easy and quick to obtain, fast to process, giving reproducible results, possible to obtain even from uncooperative patients, and, in the future, probably cheap. In this study, we showed that serum levels of MMP-2, MMP-9, and CCL-2 influence seizure count at follow-up and, as such, can be considered biomarkers of seizure prediction.

Additionally, and not less importantly, our study shows that serum levels of MMP-9, TIMP-1, MMP-2, CCL-2, and P-selectin differed even though disease activity (measured by seizure count and seizure severity) was similar. This implies that for the abovementioned molecules, it is advisable to collect blood more than once to obtain reliable results.

Finally, our knowledge of the influence of the examined molecules on seizures and epilepsy course and pathomechanisms needs further research as the present study brought not only answers but also questions.

## Figures and Tables

**Figure 1 ijms-23-14712-f001:**
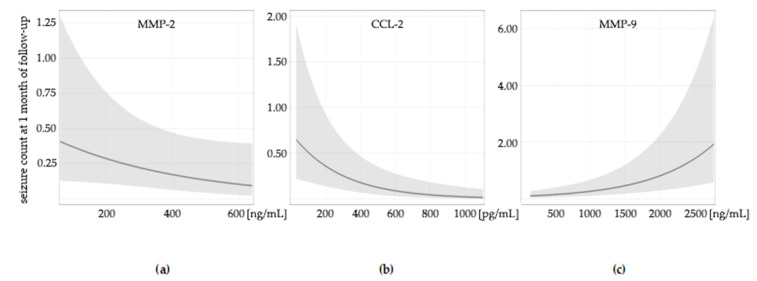
The influence of molecule levels on seizure count at one month of follow-up: (**a**) MMP-2; (**b**) CCL 2; (**c**) MMP 9. The influence of molecule levels on seizure count at seven days, 3, 6, and 12 months are shown in Appendix A.

**Figure 2 ijms-23-14712-f002:**
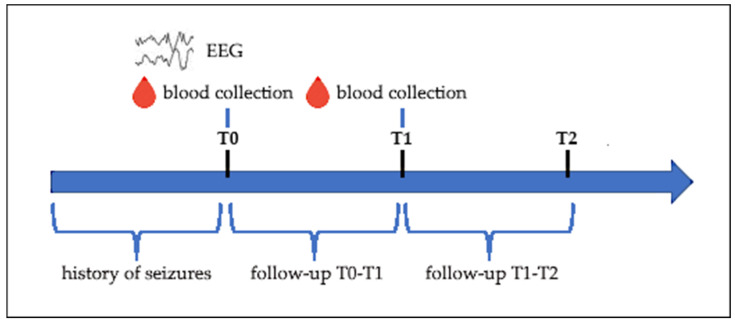
The timeline of the study.

**Table 1 ijms-23-14712-t001:** Clinical and demographic characteristics of the patients.

	Mean ± SEM	Median	Min–Max
**age (years)**	44.84 ± 2.26	42	19–82
**age at epilepsy onset (years)**	24.45 ± 2.35	21	1–67
**epilepsy duration (years)**	20.43 ± 1.90	17	2–57
**time since last seizure (days)**	607.60 ± 114.30	305	7–4036
**positive family history of epilepsy (number of patients)**		7	
**positive history of febrile seizures (number of patients)**		3	
**drug-resistance (number of patients)**		25	

**Table 2 ijms-23-14712-t002:** Clinical characteristics of the patients–etiology.

EPILEPSY ETIOLOGY	
**GENETIC EPILEPSY**	18
**idiopathic generalized epilepsy**	18
**STRUCTURAL-METABOLIC EPILEPSY**	24
**MTLE * with hippocampal sclerosis**	2
**MTLE without hippocampal sclerosis**	5
**post-traumatic epilepsy**	6
**vascular epilepsy**	5
**epilepsy due to neonatal hypoxic-ischemic encephalopathy**	2
**epilepsy due to congenital malformations of the CNS**	2
**post-infectious epilepsy**	2
**EPILEPSY OF UNKNOWN ETIOLOGY**	7

***** MTLE–mesial temporal lobe epilepsy.

**Table 3 ijms-23-14712-t003:** Clinical characteristics of the patients–seizure type.

Seizure Type		
Both Primary and Secondary Tonic-Clonic Seizures	Focal Seizures (Motor and Non-Motor)	Both Focal Seizures Secondary Generalized Tonic-Clonic Seizures
21	7	21

**Table 4 ijms-23-14712-t004:** Clinical characteristics of the patients–seizure count at baseline.

	Seizure Count at Baseline (T0)								
	History				T0-T1				
Period [m–months. d–days]	12 m	6 m	3 m	1 m	7 d	1 m	3 m	6 m	12 m
**mean** **± SEM**	6.18 ± 1.56	4.12 ± 1.13	1.65 ± 0.44	0.87 ± 0.12	0.08 ± 0.28	0.61 ± 0.28	2.36 ± 1.10	4.47 ± 1.78	6.49 ± 2.52
**median**	1	0	0	0	0	0	0	0	1
**min–max**	0–57	0–44	0–14	0–3	0–1	0–13	0–51	0–74	0–109

**Table 5 ijms-23-14712-t005:** Clinical characteristics of the patients–seizure count at follow-up.

	Seizure Count at Follow-Up (T1)								
	T0-T1				T1-T2				
Period [m–months. d–days]	12 m	6 m	3 m	1 m	7 d	1 m	3 m	6 m	12 m
**mean** **± SEM**	7.31 ± 2.54	3.71 ± 1.28	2.06 ± 0.66	0.57 ± 0.19	0.06 ± 0.05	0.61 ± 0.20	1.61 ± 0.52	3.16 ± 1.02	6.88 ± 2.10
**median**	0	0	0	0	0	0	0	0	1
**min–max**	0–108	0–54	0–25	0–7	0–2	0–7	0–21	0–42	0–75

**Table 6 ijms-23-14712-t006:** Clinical characteristics of the patients–EEG at baseline.

	True	False
**normal EEG**	18	31
**abnormal background activity**	7	42
**focal slow waves**	26	23
**epileptiform discharges**	9	40

**Table 7 ijms-23-14712-t007:** Clinical characteristics of the patients–prediction of seizure recurrence as assessed with EPT.

Feature	2-Year Seizure Recurrence Risk	5-Year Seizure Recurrence Risk	10-Year Chance of Seizure-Freedom (Seizure-Free for at Least 1 Year)
**mean** **± SEM (%)**	77.76 ± 1.48	85.60 ± 1.10	25.12 ± 1.86
**median** **(%)**	78	89	27
**min–max** **(%)**	43–90	53–90	20–60

**Table 8 ijms-23-14712-t008:** Clinical characteristics of the patients–treatment.

	Number of Patients		
	Monotherapy at T0/T1	Polytherapy at T0/T1	Total at T0/T1
**valproate**	16/14	9/10	25/24
**carbamazepine**	7/7	5/4	12/11
**lamotrigine**	6/6	5/6	11/12
**levetiracetam**	4/5	5/6	9/11
**topiramate**	0/0	3/3	3/3
**gabapentin**	0/0	2/2	2/2
**phenobarbital**	1/1	0/0	1/1
**phenytoin**	0/0	1/1	1/1
**total**	**34/32**	**15/17**	**49**

**Table 9 ijms-23-14712-t009:** Levels of the examined molecules at baseline and follow-up.

	Baseline			Follow-Up			
	Mean ± SEM	Median	Min–Max	Mean ± SEM	Median	Min–Max	*p*
**MMP-9** **[ng/mL]**	780.33 ± 72.39	601.92	125.22 –2555.34	449.79 ± 48.51	416.44	10.15–1699.80	<0.0001
**TIMP-1** **[ng/mL]**	236.83 ± 16.38	259.14	2.69–464.72	100.60 ± 7.71	100.85	3.83–216.35	<0.0001
**MMP-2** **[ng/mL]**	289.80 ± 15.35	272.59	133.86–548.59	200.84 ± 13.62	211.34	41.32–496.85	<0.0001
**CCL-2** **[pg/mL]**	325.95 ± 21.12	312.35	112.47–771.43	434.65 ± 27.60	394.29	93.14–1299.29	<0.0001
**P-sel** **[ng/mL]**	105.48 ± 9.70	91.09	6.62 –240.68	159.10 ± 15.65	135.79	16.47–389.76	0.0035
**MMP-2/TIMP-2**	1.95 ± 0.10	1.84	0.83 –3.86	1.28 ± 0.08	1.33	0.29 –3.41	<0.0001
**TIMP-2** **[ng/mL]**	149.71 ± 4.45	146.91	95.16 –257.58	159.51 ± 4.57	164.07	92.84–244.59	0.094
**S100B** **[pg/mL]**	43.32 ± 12.32	24.93	0.29 –571.13	42.34 ± 13.2	11.88	0.5 –516.25	0.2674
**ICAM-1** **[ng/mL]**	181.91 ± 9.75	163.02	103.34 –437.03	167.02 ± 8.4	155.19	36.03–369.73	0.06
**TSP-2** **[ng/mL]**	29.13 ± 2.42	25.55	5.71 –68.80	26.35 ± 1.8	22.44	11.22 –66.6	0.4062
**MMP-9/TIMP-1**	7.79 ± 2.58	2.77	0.66 –131.21	14.39 ± 4.24	6.21	0.07–299.90	0.1645

## Data Availability

The data set can be obtained at https://docs.google.com/spreadsheets/d/1HYM88Evs3kX3CNxQOyXOVGY6-lgK5Msk/edit?usp=sharing&ouid=108322422041402375271&rtpof=true&sd=true.

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
