# Peer review of "Serum Proteins Associated with Blood–Brain Barrier as Potential Biomarkers for Seizure Prediction"

_ijms, 2022, doi:10.3390/ijms232314712_

Round 1

Reviewer 1 Report

Interpretation: 

Molecules related to the BBB cannot be considered predictors of seizures. The results obtained in Table 9 show that there is no difference in the molecules evaluated between the patients who presented seizures and the patients without seizures. However, in figure 2, when making the connection between the number of crises during a month and the concentrations of the evaluated molecules MMP2, CCL-2, MMP2, they do not reflect that they are predictors of critical as commented. It cannot be concluded that MMP2, CCL-2, MMP2 as seizure predictors.

Although the design of the study and the analysis of results seem correct, the presentation of results can be significantly improved the tables. It is suggested to make a time diagram considering the EPS analysis and the taking of blood samples.

Introduction

Line 44-46, the literature (8,3) does not adequately support the relationship between inflammation and BBB activation associated with exacerbation of seizures

Line 48- the authors do not support bibliographically why they chose the molecules to be evaluated...nor the relationship in the regulation of BBB homeostasis and inflammation.

Results 

Line 90. It is suggested to improve the quality of table 9. Properly separate the numbers

The authors mention the seizure count in the period of last 1, 3, 6, and 12 months were taken at baseline (T0). But they do not make clear how long the blood samples were taken. If they took a single blood sample at baseline and at follow-up, because in the method mentioned …After blood collection the 30 minute EEG examination in 10-20 montage was performed. Clarify

Author Response

Dear Reviewer,

Thank you very much for reviewing our paper and providing valuable comments. We carefully considered your suggestions and tried our best to address them to improve the manuscript. We hope that the revised version of the manuscript meets your high standards. We provide the point-by-point responses below. We kept the revised version of the manuscript in a review mode so that the corrections are easier to follow (with a minor exception of Table 9 as it was pointed out in Response 5.).

We welcome further constructive comments.

Kind regards,

Elzbieta Bronisz

  1. Molecules related to the BBB cannot be considered predictors of seizures. The results obtained in Table 9 show that there is no difference in the molecules evaluated between the patients who presented seizures and the patients without seizures. However, in figure 2, when making the connection between the number of crises during a month and the concentrations of the evaluated molecules MMP2, CCL-2, MMP2, they do not reflect that they are predictors of critical as commented. It cannot be concluded that MMP2, CCL-2, MMP2 as seizure predictors.

Response 1:

In line with your suggestions we revised the paragraph on blood-brain barrier and its connection to seizures and epilepsy to improve the introduction (lines 49-69).

We apologize for the description of methods and presentation of results as they might have been unclear. We consulted statistical analyst to re-analyze the data and help with describing the methods in a more precise way. In Table S1 we showed that there were no significant differences in the levels of biomarkers between the no-seizure and seizure groups at baseline. Yet in the GLMMs both group of patients, with and without seizures, were analyzed to check if the serum levels of molecules has influence on seizure count. In this analysis we concentrated not only on the sole presence of seizures or lack of them, but we looked at seizure count during the follow-up. The GLMMs that were used in our study allow for eliminating biases resulting from the ICC (Interclass Correlation) autocorrelation between the measurements. We used Akaike Information Criterion to select the optimal model. AIC statistics are asymptotically equivalent to resampling methods. [Konishi S. et al., "Information Criteria and Statistical Modeling", Springer 2008, pp. 245-247 (Asymptotic Equivalence Between AIC-Type Criteria and Cross-Validation)]. In cases where there was an excess of “zero” responses (over 10%), the Tweedie distribution was used (the Tweedie distribution is suitable for data with a zero type object called a “point mass”). In other cases the distributions were selected from the class of the exponential family .

  1. Although the design of the study and the analysis of results seem correct, the presentation of results can be significantly improved the tables. It is suggested to make a time diagram considering the EPS analysis and the taking of blood samples.

Response 2:

Thank you for pointing this out to us. We designed a time diagram (Figure 2; lines 659-664) with time points of blood collection, EEG examination and follow-up.

  1. Introduction

Line 44-46, the literature (8,3) does not adequately support the relationship between inflammation and BBB activation associated with exacerbation of seizures

Response 3:

We cited more literature supporting the relationship between inflammation, BBB activation and exacerbation of seizures (lines 46-48).

  1. Line 48- the authors do not support bibliographically why they chose the molecules to be evaluated...nor the relationship in the regulation of BBB homeostasis and inflammation.

Response 4:

We corrected this paragraph and cited the appropriate literature according to the suggestions (lines 49-72).

  1. Results 

Line 90. It is suggested to improve the quality of table 9. Properly separate the numbers

Response 5:

Thank you for pointing that to us. We corrected the table (line 112). As it was difficult to track the changes in the review mode, the columns that had been corrected are in red.

  1. The authors mention the seizure count in the period of last 1, 3, 6, and 12 months were taken at baseline (T0). But they do not make clear how long the blood samples were taken. If they took a single blood sample at baseline and at follow-up, because in the method mentioned

Response 6:

Thank you for your suggestion, we designed a time diagram (Figure 2; lines 659-664) to facilitate understanding of the study design. We counted seizures in the period of last 1, 3, 6, and 12 months at T0 and the blood was collected at T0. Additionally, we asked our patients to write down their seizures for a period of one year until T1 and we counted seizures at 7 days, 1, 3, 6, and 12 months from T0. At T1, we again counted seizures in the period of last 1, 3, 6, and 12 months from T1 and again the blood was collected. The patients were followed-up for one year and we obtained seizure count at at 7 days, 1, 3, 6, and 12 months from T1. The blood collection took place once at T0 and once at T1. Only after statistical analysis which took place after T2 we got to know that the levels of the biomarkers had changed.

  1. …After blood collection the 30 minute EEG examination in 10-20 montage was performed. Clarify

Response 7:

At T0 we performed standard EEG using 10-20 montage and the examination lasted 30 minutes. We corrected the paragraph to make it clearer to the readers (lines 622-623).

Reviewer 2 Report

The authors measure serum protein levels in blood samples of chronic epileptic patients. The type of epilepsy, the number of seizures during the experimental period and the medication (types of medicaments used) were detected. The authors aimed the prediction of seizures on the basis of the serum protein levels. The proteins measured were inflammation markers and proteins related to BBB function. They found three molecules (MMP-2; CCL-2; MMP-9) which were related to the number of seizures counted by the patients.

Weak points are as follows: 1. Figure 1 is completely unnecessary, because the text indicates the facts of the flow chart clearly. 2. The discussion is too long; we think that only those molecules should be discussed which presented correlation to the number of the seizures. Therefore the discussion of the role of ICAM-1, S100B, TIMP-1, TIMP-2 and TSP-2 should not be discussed. Also the section 3.10. Co-variables could have been deleted. This way the article will be shorter, but with more compact information content.

Author Response

Dear Reviewer,

Thank you very much for reviewing our paper and providing valuable comments. We carefully considered your suggestions and tried our best to address them to improve the manuscript. We significantly abbreviated the part of discussion concerning the molecules that were not predicting seizure count after examination. We thoroughly considered the paragraph concerning clinical co-variables, yet both clinical and statistical importance decided against removing it. We provide broader responses below. We hope that the revised version of the manuscript meets your high standards.

We welcome further constructive comments. The revised version of the manuscript is in a review mode so that the corrections are easier to follow.

Kind regards,

Elzbieta Bronisz

The authors measure serum protein levels in blood samples of chronic epileptic patients. The type of epilepsy, the number of seizures during the experimental period and the medication (types of medicaments used) were detected. The authors aimed the prediction of seizures on the basis of the serum protein levels. The proteins measured were inflammation markers and proteins related to BBB function. They found three molecules (MMP-2; CCL-2; MMP-9) which were related to the number of seizures counted by the patients.

Weak points are as follows:

  1. Figure 1 is completely unnecessary, because the text indicates the facts of the flow chart clearly.

Response 1:

Thank you for indicating that. We moved the flow chart to supplementary materials (Figure S1).

  1. The discussion is too long; we think that only those molecules should be discussed which presented correlation to the number of the seizures. Therefore the discussion of the role of ICAM-1, S100B, TIMP-1, TIMP-2 and TSP-2 should not be discussed. Also the section 3.10. Co-variables could have been deleted. This way the article will be shorter, but with more compact information content.

Response 2:

Thank you for pointing out the length of discussion to us. We significantly abbreviated the part of discussion about the role of ICAM-1, S100B, TIMP-1, TIMP-2 and TSP-2 fitting it into one subchapter (lines 289-318) as we believe that mentioning them allows for a better understanding of possible underlying processes and does not elongate the article too much, but resigned from assigning one subchapter to each molecule. As far as the subchapter about the co-variables is concerned, this data concerns clinically and statistically important factors. Time of remission or disease duration are factors known to influence seizure incidence and as such cannot be omitted from the clinical point of view. Moreover, the GLMMs indicate their impact on seizure count together with the level of MMP-2, MMP-9, CCL-2 and not including them into the discussion could be leading to an unnecessary bias.

Reviewer 3 Report

The manuscript investigates an interesting and timely question in molecular neuroscience and epilepsy research. However, insufficient information regarding the statistical tests performed was included and in fact, I’m not entirely convinced that the statistical results were robust given the embedded statistical biases within the data. I suggest that the authors carefully analyze the data preferably by using non-parametric/resampling methods to ensure such biases would not influence the observed findings.

Author Response

Dear Reviewer,

Thank you very much for reviewing our paper and providing valuable comments. We carefully considered your suggestions and tried our best to address them to improve the manuscript. We corrected the description of the statistical analysis and re-checked the results. We hope that the revised version of the manuscript meets your high standards. We provide the response below.

We welcome further constructive comments.

The revised manuscript is in a review mode so that the corrections are easier to follow (with a minor exception of Table 9 as the corrected columns are in red).

Kind regards,

Elzbieta Bronisz

The manuscript investigates an interesting and timely question in molecular neuroscience and epilepsy research. However, insufficient information regarding the statistical tests performed was included and in fact, I’m not entirely convinced that the statistical results were robust given the embedded statistical biases within the data. I suggest that the authors carefully analyze the data preferably by using non-parametric/resampling methods to ensure such biases would not influence the observed findings.

Answer:

Thank you very much for your review. In line with your suggestions we revised the paragraph on blood-brain barrier and its connection to seizures and epilepsy to improve the introduction (lines 49-69). We apologize for the description of methods and presentation of results as they might have been unclear. We consulted statistical analyst to re-analyze the data and help with describing the methods in a more precise way (lines 640-657).

Statistical biases embedded within the data may be related to a patient selection. The article describes an observational study and the possible problems could relate to the method of selecting patients - however, efforts were made to avoid this through an appropriate design of the study (description is beyond the scope of this article).

The possibility of applying adjustments such as a propensity score has been taken into account. Unfortunately, this allowed to control only some of the variables and hidden biases could remain after adjustment. It was decided that the reduction of the bias could be provided by modeling "causal" relationships between the treatment and observed / unobserved covariates directly using GLM / GLMM models [Valliant R. et al,” Practical Tools for Designing and Weighting Survey Samples” Springer 2018].  

The impact of both the probability distributions and the structure of risk factors on the mean values of  the targets was examined. Tests based on the AIC (Akaike Information Criterion) criterion were used to select the optimal model. AIC statistics are asymptotically equivalent to resampling methods. [Konishi S. et al., "Information Criteria and Statistical Modeling", Springer 2008, pp. 245-247 (Asymptotic Equivalence Between AIC-Type Criteria and Cross-Validation)]. GLMM models also allow for eliminating biases resulting from the ICC (Interclass Correlation) autocorrelation between the measurements.

In cases where there was an excess of “zero” responses (over 10%), the Tweedie distribution was used (the Tweedie distribution is suitable for data with a zero type object called a “point mass”). In other cases the distributions were selected from the class of the exponential family .

Round 2

Reviewer 3 Report

I still believe that the statistical analyses performed are not sufficiently justified and described. Please at least address the limitations of the selected methods in the discussion to help readers interpret the findings.

Author Response

Dear Reviewer,

Thank you very much for your hard work reviewing our paper and providing your suggestions. We consulted the statistician and added a paragraph addressing the fact that the methods that were used have limitations (Discussion, lines 381-385). We also revised the description of the statistical methods (Materials and Methods, lines 441-442). After careful consideration we decided against long and thorough discussion of the limitations of the statistical methods as the reader would probably be more interested in the molecules and their possible use than in a long discussion which could be found in specialistic statistical literature, yet we included references to facilitate and encourage further reading. We hope that the revised version of the manuscript meets your high standards. We provide the added lines and a more detailed answer below.

Thank you very much for your time and helping us to improve the manuscript.

We welcome further constructive comments.

Kind regards,

Elzbieta Bronisz

I still believe that the statistical analyses performed are not sufficiently justified and described. Please at least address the limitations of the selected methods in the discussion to help readers interpret the findings.

Answer:

In the analysis of this study, some models from the class of GLMM (Generalized Linear Mixed Models) were used. Of course, they have their limitations, but they are commonly used in practice to describe studies similar to those in this manuscript. It is difficult to describe the details of the statistical methodology in a short article but we will be happy to answer all specific questions. Thank you for comments about the data selection and that “the statistical results should be robust given the embedded statistical biases within the data”. The suggestions regarding the “use of the non-parametric / resampling methods to ensure such biases would not influence the observed findings” are very interesting, but the literature refers rather to the parametric approach used in this study.

Much more often, instead of using robust or non-parametric methods, broad classes of probability distributions are used. Paul Allison states that robust methods, used e.g. in the imputation of missing data, excessively reduces the variance of the estimators. Monte Carlo studies have shown that the first and second order bias corrections in resampling methods are superior to the corresponding analytical methods, but on the other hand we encounter new problems due to sampling fluctuations. Nevertheless, we thank you for your critical comments.

Of course, as part of the analysis, many other calculations were made (without citing their results) in the search of the optimal solution. However, the work was primarily concerned with the medical problems and therefore the statistical discussion was neglected. When building the models, we relied on the literature below:

  • by R. L. Chambers, C. J. Skinner, „Analysis of Survey Data”, 2003, Wiley, (pp 29-48, 175-194, 289-306).
  • Cleophas T., Zwinderman A., „Statistics Applied to Clinical Studies”, 2012, Springer (5ed), (pp 119-130, 177-185, 329-336).
  • Brown H., Prescott R., „Applied Mixed Models in Medicine”, 2015, Wiley (3ed), (pp 19-31, 34-53, 69-88, 125-142, 231-288).
  • Rosenbaum P. „Design of Observational Studies”, 2010, Springer, (pp 95-110, 257-272, 287-298).
  • Rassler S. „Statistical Matching. A Frequentist Theory, Practical Applications, and Alternative Bayesian Approaches”, 2002 Springer, (pp 44-69, 128-197).
  • Frees E., „Regression Modeling with Actuarial and Financial Applications”, 2010, Cambridge University Press, (pp 148-182, 362-378, 433-451).
  • Hastie, T. Tibshirani, R. and Friedman, J., „The Elements of Statistical Learning: Data Mining, Inference, and Prediction”, 2008, Springer (2ed), (pp 43-94, 219-257, 261-293, 389-415).

Added lines (381-385):

The statistical analysis involved the models from the class of GLMMs and as with every statistical method, the GLMMs have their limitations [109]. In the light of these limitations we carefully selected models from the class of GLMMs using the practical information-theoretic approach [110] to avoid misselection. A more complex discussion on the limitations of the GLMMs is beyond the scope of this article, yet it could be found in [109].

Added lines (441-442):

Models were selected using the practical information-theoretic approach [110].
